# The Infectious Clue: Linking Bacterial Infections to Underlying Malignancies

**DOI:** 10.3390/cancers17243958

**Published:** 2025-12-11

**Authors:** Loris Močibob, Branimir Gjurašin, Neven Papić

**Affiliations:** 1Department of Gastrointestinal Infections, University Hospital for Infectious Diseases Zagreb, 10000 Zagreb, Croatia; lmocibob@bfm.hr; 2School of Medicine, University of Zagreb, 10000 Zagreb, Croatia; bgjurasin@bfm.hr; 3Department of Intensive Care Medicine, University Hospital for Infectious Diseases Zagreb, 10000 Zagreb, Croatia

**Keywords:** occult malignancy, sentinel infections, bacteremia, anaerobic bacteria, dysbiosis, colorectal cancer, hepatobiliary cancer, *Streptococcus gallolyticus*, *Fusobacterium*, early cancer detection

## Abstract

A blood culture result indicating a hidden tumor is a crucial scenario that highlights the need to reconsider the clinical significance of some infections traditionally viewed as isolated events. Recent research demonstrates that certain infections may serve as early indicators of occult malignancies. Specific infection patterns, such as unexplained liver abscesses, persistent pneumonia, or bloodstream infections with gut bacteria, frequently precede cancer diagnosis by several months. These events are not random. Tumors can compromise immune function, disrupt tissue barriers, and alter the microbiome, thereby facilitating bacterial infections. Recognizing these infection patterns challenges conventional clinical approaches and prompts the critical question of whether cancer should be sought when infections lack an obvious cause. Identifying this association may enable earlier cancer detection, reduce the incidence of advanced-stage diagnoses, and improve patient survival.

## 1. Introduction

A 68-year-old male patient presented with *Escherichia coli* bacteremia. No identifiable infectious focus was found, and inflammatory markers normalized rapidly with antibiotic therapy, leading to a diagnosis of “transient gut translocation”. Six weeks later, he was re-admitted with *Enterobacter* spp. bacteremia. This time computed tomography and colonoscopy identified a stage II sigmoid adenocarcinoma. Although such scenarios are routinely encountered in hospitals globally, the oncologic significance is often initially overlooked, highlighting a critical knowledge gap—when should unexplained Gram-negative bacteremia prompt targeted evaluation for occult malignancy?

Global cancer incidence is increasing, with 19.3 million new cases reported in 2020 and projections of up to 30 million annually by 2040 [1]. For most solid tumors, delayed diagnosis remains the main barrier to better outcomes. Only a minority of colorectal (CRC) and lung cancers are found through screening—up to 70% are diagnosed at stage III–IV, with nearly half of lung cancers already metastatic [2,3,4].

The link between chronic inflammation and cancer is not a modern concept. In early Chinese medical texts, Galen’s humoral pathology and Virchow’s “chronic irritation theory”, persistent inflammation was linked with “mass-forming disease”. Modern infectious diseases re-exposed this link in a clinically actionable way. In 1951, McCoy and Mason described an association of enterococcal endocarditis with CRC [5], raising the concept that CRC can serve as a portal of entry for gut bacteria, which was later confirmed by Klein et al. in 1977 leading to current recommendations for colonoscopic evaluation in patients with *Streptococcus bovis* bacteremia or endocarditis [6].

Since then, multiple studies have shown that certain infections are followed by sharply increased short-term cancer detection. For example, overall cancer incidence during the first 6 months after Gram-negative bacteremia is about threefold higher than expected, after pneumonia 2.5-fold higher, and in patients with pyogenic liver abscess (PLA) the risk of subsequently diagnosed cancer is increased by fourfold, with an especially high excess risk in the first few months after the infection episode [7,8,9,10,11,12,13]. This highlights that infection is not generating cancer—it can be the first visible sign of a malignancy that has been present silently for years.

Despite substantial evidence, this association is not consistently recognized in clinical practice. The segmentation of contemporary medicine is often the reason why no single specialty oversees the entire diagnostic process from the initial infectious event to the eventual identification of occult malignancy. Consequently, the oncologic significance of certain infections is frequently overlooked or recognized too late. Furthermore, very few reports synthesize epidemiologic data, mechanistic immunobiology, and diagnostic implications into a coherent clinically applicable framework.

This review uniquely integrates epidemiologic evidence, immunobiology, and bedside algorithms to demonstrate that specific infections may serve as early indicators of occult malignancy. It examines the biological mechanisms through which tumors predispose individuals to infection and proposes a practical, pathogen-specific diagnostic framework termed the “sentinel infection phenotype”. In defined clinical contexts, infections should be viewed not only as isolated events but also as potential indicators of underlying malignancy.

## 2. Infections Forecasting Cancer: What the Numbers Reveal

Recent studies show that certain bacterial infections often occur shortly before the diagnosis of solid cancers and that this timing pattern is replicable across different populations. Multiple epidemiological studies showed that the excess cancer risk following infection is highest in the first months, begins to decline by about 6 months, and continues to attenuate thereafter [8,13,14]. In general populations, the absolute risk of being diagnosed with cancer in the first six months after a bacterial infection is usually around 1–3%, with standardized incidence ratios (SIRs) of 2 to 4 for any cancer type during this period [13,15]. Higher detection rates are seen in certain clinical subgroups, particularly older adults with unexplained bacteremia and smokers who present with lobar pneumonia [10,16]. This temporal pattern supports a reverse-causation model, in which the infection is often the first clinical manifestation of an already existing but previously undiagnosed tumor, rather than a distant causal factor in cancer development. Some of this early excess in cancer diagnoses is likely driven by greater healthcare contact and more intensive diagnostic testing around the index infection; however, the magnitude and consistent, organ-specific clustering of risk across cohorts suggests that detection bias alone is unlikely to fully explain these associations.

Beyond this short-term diagnostic window, a second pattern is apparent: several chronic colonization or infection states (for example chronic *Salmonella Typhi* carriage, recurrent urinary tract infections, or persistent *Chlamydia pneumoniae* seropositivity) have been associated with a modest but sustained increase in long-term cancer risk, probably mediated through chronic inflammation, genotoxic effects, and pro-oncogenic changes in the local microbiome. The strength of evidence for these infection–cancer links is heterogeneous, ranging from well-established associations supported by large population-based cohorts (e.g., *Streptococcus gallolyticus* and colorectal cancer) to emerging or largely anecdotal signals. Importantly, most of the evidence summarized in this section comes from retrospective observational studies, which are inherently vulnerable to confounding, detection bias and surveillance bias; these methodological limitations are discussed in more detail in Section 7.

### 2.1. Gastrointestinal Cancers (Colorectal and Gastric)

CRC shows the strongest and most clinically relevant association with infection-related warning signs (Table 1). National Danish registry data show that among more than 11,000 patients with Gram-negative bacteremia, about 3% are diagnosed with cancer within six months, with gastrointestinal cancers occurring 3- to 13-times more often during this period [13]. A recent population-based study from Queensland further confirmed this pattern, demonstrating elevated short-term CRC detection following bloodstream infection, particularly after anaerobic and Gram-negative episodes [14]. Pathogen-specific cohort analyses confirm that bacteremia with *Fusobacterium nucleatum*, *Bacteroides* and *Clostridium septicum* is associated with a markedly increased short-term risk of CRC (with adjusted risks that are several-fold higher than for other bacteremias and exceeding ten-fold for *C. septicum*) [9,17,18,19]. In patients with *Enterococcus* endocarditis, colorectal neoplasia was found in over half of cases, similar to *Streptococcus gallolyticus*, supporting the role of routine colonoscopy as part of diagnostic evaluation [20,21,22]. A large Danish study recently confirmed that patients with *Enterococcus faecalis* bloodstream infection have a significantly higher short-term risk of CRC, with a 3.7-fold increased hazard compared to matched population controls [23]. Patients with PLA, especially caused by *Klebsiella pneumoniae*, are also at higher risk of having an underlying CRC, and colonoscopy in those patients should be considered as well [11,24,25]. Among these, the associations of *S. gallolyticus* and *C. septicum* with CRC can be considered high-certainty sentinel links, whereas the signals involving *F. nucleatum*, *Bacteroides* spp. and *Enterococcus* spp. are strong but still emerging and will benefit from further prospective validation.

In addition to these sentinel infections, some chronic infection states may increase the long-term risk of cancer development. For example, *Helicobacter pylori* is a well-established long-latency carcinogen. Prospective studies and meta-analyses show that *H. pylori* infection increases gastric cancer risk by two- to threefold, with the risk rising more than sixfold when the infection is acquired early in life [26]. In regions with high prevalence, population-level eradication programs have led to measurable declines in gastric cancer incidence, further supporting a causal relationship [27]. 

### 2.2. Pancreaticobiliary Cancers

A Taiwanese study described that more than 2% of patients with PLA presented with the disease as the initial manifestation of underlying hepatocellular carcinoma (HCC) [28]. Similar patterns have been described for biliary tract cancers presenting with cholangitis or liver abscess due to tumor-related ductal obstruction.

Beyond these sentinel presentations, chronic colonization states and dysbiosis appear to influence long-term pancreaticobiliary cancer risk. Chronic *Salmonella Typhi* carriage in typhoid-endemic areas is linked to a four- to fivefold higher risk of gallbladder cancer, especially in individuals who also have gallstones [29]. Pancreatic cancer shows an association with oral dysbiosis. In a case–control study, higher pancreatic cancer risk was observed in individuals with increased oral levels of *Porphyromonas gingivalis* (OR 1.60) and *Aggregatibacter actinomycetemcomitans* (OR 2.20) [30].

### 2.3. Lung Cancer and Pulmonary Infection Phenotypes

Pulmonary infections can serve as early warning signs of hidden lung cancer. Danish national data show a 1.4% risk of being diagnosed with lung cancer within six months after hospitalization for pneumonia, with SIR around 2.5 and an eightfold excess among patients who eventually develop lung cancer [12]. In high-risk groups, especially heavy smokers hospitalized with pneumonia, the twelve-month lung cancer risk can reach 8%, and up to 24% when pneumonia affects the upper lobe [16]. This pattern is consistent with post-obstructive infection caused by tumor-related airway obstruction [16].

In parallel, chronic *Chlamydia pneumoniae* infection contribute to long-term lung cancer risk, with meta-analyses showing overall OR of 1.5 and subgroup risks up to 2.3 in IgA-positive individuals, likely driven by chronic inflammation [31].

### 2.4. Genitourinary Cancers

Strong links between urinary tract infections (UTIs) and bladder cancer are seen in large population studies. In a Taiwanese cohort of more than 70,000 UTI patients, the adjusted hazard ratios (aHR) for urinary tract cancers ranged from 4.5 to 4.7, with bladder cancer being the most common and the highest risk observed in people with repeated, unexplained infections [32]. In the Nijmegen bladder cancer study recurrent UTIs were linked to a higher risk of bladder cancer, with the strongest association seen in people who regularly experience episodes of cystitis (OR 6.6 in men and 2.7 in women), with strong association in men with more than ten antibiotic-treated UTIs (OR 6.0) [33].

Recurrent UTIs were also linked with prostate cancer in several cohorts. In an Italian case–control study cystitis was linked to a higher risk of prostate cancer (OR 1.76), with a markedly stronger association when infection occurred within one year before diagnosis (OR 7.6) [34]. In a nationwide Swedish cohort of over 600,000 adults aged 50 years and older, acute cystitis was also shown to precede urogenital cancers, with the highest risks observed within three months after infection and the reported SIR for prostate cancer of 7.1 [35]. Similarly, in Taiwanese study men with lower UTIs had a significantly higher risk of developing prostate cancer (aHR of 1.46 to 1.72) and the association was particularly strong among patients with recurrent infections, whose prostate cancer risk increased more than ninefold [36].

A meta-analysis including 47 studies found that men with a history of sexually transmitted infections had a 49% higher risk of prostate cancer (SIR 1.5), with a particularly consistent association for gonorrhea (SIR 1.20) [37]. In another meta-analysis *Chlamydia trachomatis* infection has been associated with an increased risk of cervical cancer (OR 2.2), and co-infection with human papillomavirus further amplifies this risk [38].

### 2.5. Other Pathogens and Malignancies

Frequent community-acquired infections such as bronchitis, sinusitis and pneumonia have been associated with a higher risk of subsequent multiple myeloma diagnosis [39]. *Staphylococcus aureus* bacteremia showed a higher incidence of malignancy during the first year, with the highest risks observed for cervical cancer, hematologic malignancy, sarcoma, liver and pancreatic carcinoma, and urinary tract carcinoma [8].

A variety of other bacterial pathogens have been reported preceding cancer detection; however, data are limited, and associations were largely anecdotal, based mainly on case reports and small series, and should be interpreted with caution. These include organisms such as the *Streptococcus anginosus* group, *Aeromonas* spp., *Clostridium perfringens*, *Actinomyces* spp., and *Listeria monocytogenes* [40,41,42,43,44,45], as further described.

**Table 1 cancers-17-03958-t001:** Pathogen-specific infection phenotypes and their association with subsequent cancer detection: a summary of selected published studies.

First Author, Year	Infection Phenotype	Study Design	Time Window	Effect Size/Absolute Risk
Colorectal and other gastrointestinal cancers				
Østergaard, 2025 [23]	*Enterococcus faecalis* bacteremia	Danish nationwide registries	First 6 months	CRC 0.45%, CRN 2.3%; CRC HR 3.7; CRN HR 4.6
Ursi, 2021 [22]	*Enterococcus* spp. and *S. gallolyticus* IE	Single-center cohort	At time of IE work-up	In patients with colonoscopy: 71–83% intestinal disease; 13–15% CRC
Abe, 2024 [19]	Anaerobic bacteremia	Single-center cohort	Within 1 year	OR 3.44 for any GI cancer
Justesen, 2022 [46]	Anaerobic bacteremia	Population-based cohort	Within 1 year	*C. septicum* HR 76; *Bacteroides* HR 5.95; GPAC ~11; *Fusobacterium* ~8.5
Kwong, 2018 [17]	Anaerobic bacteremia	Population-based cohort	Within 1 year	Pathogen-specific HRs: *B. fragilis* ~3.8; *S. gallolyticus* ~5.7; *F. nucleatum* ~6.9; *C. septicum* ~17
Tsai, 2016 [21]	*Streptococcus bovis* bacteremia	Single-center cohort	No defined time window	CRC detected in 30.7% of patients with *S. bovis* bacteremia
Laupland, 2023 [14]	Community-onset bacteremia	Population-based surveillance cohort	Within 1 year	Overall IRR 16; RR *C. septicum* 25, *B. ovatus* 11.8, *C. paraputrificum* 11.4, *S. infantarius* 10.6, *G. morbillorum* 6.5
Gaab, 2023 [18]	Colibactin-producing pks^+^ *Escherichia coli*	Systematic review and meta-analysis	No defined time window	Overall OR 2.3; Western countries OR 2.3; tissue-based studies OR 2.2
Huang, 2012 [24]	Pyogenic liver abscess	Retrospective cohort	Highest risk within first 2 years	CRC 2.3%; SIR 4.00; *Klebsiella* spp. PLA SIR 5.8; aHR 2.7
Suzuki, 2023 [25]	Pyogenic liver abscess	Retrospective cohort	Highest risk within 3 years after PLA	CRC 1.9% vs. 0.8%; time-dependent HRs: 3.6 (0.5 years), 2.5 (1 yr), 1.7 (2 yr), 1.4 (3 yr)
Hepatobiliary cancers (liver and gallbladder)				
Lin, 2011 [28]	Pyogenic liver abscess	Nationwide retrospective cohort	HCC diagnosed within 60 days of PLA	HCC in 2.1%; in liver cirrhosis (OR 5.1), HBV (OR 3.8), HCV (OR 3.5)
Koshiol, 2016 [29]	Chronic *Salmonella Typhi* carriage	Case–control studies/meta-analysis	No defined time window	High Vi antibody titer OR ≈ 4.0; meta-analysis summary RR 4.6–5.0 for *S. Typhi* and gallbladder cancer
Lung cancer				
Zhan, 2011 [31]	*C. pneumoniae* infection	Meta-analysis	No defined time window	Overall OR 1.5; prospective OR 1.2; retrospective OR 2.2; IgA ≥ 64 OR 2.4
Shepshelovich, 2016 [16]	Pneumonia in smokers	Retrospective cohort	1-year follow-up	1-year lung cancer 8.1% (≈24% after upper-lobe pneumonia 76% located in same lobe)
Urinary tract cancers				
Sun, 2013 [32]	Urinary tract infection	Nationwide cohort	Risk highest in first 4 years	Any UTI HR 4.7; upper UTI HR 4.3 for renal pelvis/ureter cancer; lower UTI HR 5.7 for bladder cancer.
Vermeulen, 2015 [33]	Recurrent cystitis	Case–control study	No defined time window	Recurrent cystitis: OR 6.6 in men, OR 2.7 in women. Frequent recurrence (>10 episodes) in men: OR ~6
Fan, 2017 [36]	Lower urinary tract infection	Nationwide population-based cohort	Follow-up for up to 14 yrs	Prostate cancer higher in cystitis aHR 1.5 and urethritis aHR 1.7 vs. no UTI; >5 LUTI visits/yr → aHR 9.3
Zhu, 2016 [38]	*Chlamydia trachomatis* infection	Systematic review and meta-analysis	No defined time window	Overall OR ≈ 2.2 for cervical cancer; HPV + *C. trachomatis* OR ≈ 4.0
Any cancer detection—multiple-site signal after severe infection				
Søgaard, 2020 [15]	Community-acquired *Escherichia coli* bacteremia (age ≥ 50)	Population-based cohort	Strongest association within 1 year	1-year cancer incidence 3.0% (GI/hepatobiliary 1.9%, urinary 1.0%) SIR < 1 yr: GI/hepatobiliary 5.4; CRC 4.4; pancreas 7.2; kidney 10.5 SIR ≥ 1 yr: CRC 1.4; pancreas 2.3; overall 1.3
Søgaard, 2017 [13]	First-time Gram-negative bacteremia	Nationwide cohort	0–6 and 6–12 months after bacteremia	Overall SIR ≈ 1.4. Any cancer SIR ≈ 3.3 in first 6 months; particularly high SIRs (>4–10) for GI and GU cancers
Thomsen, 2013 [20]	Infective endocarditis	Retrospective cohort	Very high risk 0–3 months; persistent 3 months–5 yrs; modest > 5 yrs	Overall SIR 1.6: 0–3 months SIR 8; 3 months–5 yrs SIR 1.5; >5 yrs SIR 1.2 Site-specific SIR < 3 months: colon ~12, liver ~46, hematologic ~24
Gotland, 2020 [8]	*Staphylococcus aureus* bacteremia	Nationwide matched cohort study	Within first year	1-year IRR 1.65; highest site-specific IRRs: cervical 37.8, myeloma 6.3, leukemia 4.7, sarcoma 4.7, liver 3.6, pancreas 2.8, urinary 2.6.
McShane, 2014 [39]	Common community-acquired infections	Population-based case–control study	>13 months prior to diagnosis	ORs 1.1–1.4 for significant infections (strongest: pneumonia OR 1.3; sinusitis OR 1.15; bronchitis OR 1.14) Associations persist up to >72 months

Effect sizes are reported as the hazard ratio (HR), adjusted hazard ratio (aHR), odds ratio (OR), relative risk (RR), incidence rate ratio (IRR), or standardized incidence ratio (SIR) as provided by each study. Time windows refer to the period between infection episode and cancer diagnosis. Abbreviations: CRC—colorectal cancer; CRN—colorectal neoplasia; GI—gastrointestinal; GPAC—Gram-positive anaerobic cocci.

## 3. Infection-Cancer Nexus: A Pathogen Map

The infection–cancer signal is not random. Across studies, distinct microbial niches map to specific malignancies. First, gut-associated anaerobes consistently cluster with colorectal neoplasia: *Streptococcus gallolyticus*, *Clostridium septicum*, *Fusobacterium nucleatum*, enterotoxigenic *Bacteroides fragilis*, and pks^+^ *Escherichia coli* are all enriched at or within colorectal tumors and can adhere to or invade dysplastic epithelium while expressing virulence factors (e.g., colibactin, FadA adhesin, *B. fragilis* toxin) that induce DNA double-strand breaks, activate β-catenin/NF-κB/IL-17–STAT3 signaling, and drive epithelial proliferation and inflammation [14,17,18,47,48]. Emerging data also implicate *Enterococcus faecalis* bacteremia and *Streptococcus anginosus* group infections, which frequently arise from mucosal barrier breach, as a possible signal of colorectal neoplasia [14,17,19,22].

Second, hepatobiliary pathogens appear to be linked to pancreaticobiliary cancers. *Klebsiella pneumoniae* in cryptogenic PLA repeatedly points to occult colon or biliary tract neoplasm, while chronic *Salmonella Typhi* carriage remains one of the strongest region-specific associations with gallbladder cancer [24,25,29]. *Aeromonas* spp. bacteremia is likewise enriched in patients with hepatobiliary malignancy, although evidence is limited to smaller clinical studies [41].

Third, respiratory pathogens and airway dysbiosis may be associated with lung cancer. While pneumonia in smokers (especially of the upper-lobe) can unmask obstructive lung tumors, *Chlamydia pneumoniae* seropositivity—particularly elevated or persistent IgA titers—has been associated with a modest increase in lung cancer risk, especially in smokers [16,49]. Beyond classical infections, bronchoalveolar lavage and lung-tissue microbiome studies show that lung cancer samples are disproportionately populated by oral commensal/anaerobic genera—including *Veillonella*, *Prevotella*, *Fusobacterium*, *Porphyromonas* and related taxa—consistent with chronic microaspiration of the oropharyngeal microbiota [50,51,52].

In addition to some well-described pathogen–cancer relations, several organisms show weak or anecdotal associations that may reflect mucosal disruption, immune compromise, or incidental detection. *Staphylococcus aureus* bacteremia is followed by a short-term diagnostic spike in several malignancies [8]. *Clostridium perfringens* sepsis, pelvic *Actinomyces* infection, and invasive *Listeria monocytogenes* disease have each been described in patients later diagnosed with gastrointestinal malignancy [40,41,42,43,44,45]. However, *Listeria* infections more commonly occur in individuals with previously diagnosed malignancy [53], whereas *Actinomyces* infections typically mimic malignant disease rather than precede it [54,55]. A comprehensive organism–cancer–mechanism map is shown in Figure 1.

## 4. Mechanistic Insights: How Tumors Enable Infection

Even clinically or radiologically invisible cancers can act like silent saboteurs of the immune system, possibly leaving the host vulnerable to infections. Mechanistically, early tumors induce (i) systemic immunosuppression, (ii) local ecological changes in the tumor microenvironment (TME) that enable microbial persistence, (iii) gut and mucosal dysbiosis with enrichment of pro-oncogenic and invasive taxa, creating (iv) chronic inflammatory feedback loops that might accelerate malignant evolution while progressively degrading host defense (Figure 2).

### 4.1. Early Systemic Immune Reprogramming in Cancer—Relevance for Infection Biology

Clinically relevant immunosuppression is not confined to late-stage malignancy. Transcriptomic profiling consistently shows that even early or pre-malignant solid tumors, such as colorectal advanced adenomas, induce systemic immune reprogramming—innate immune activation followed by T-cell suppression—a feature that is detectable in circulating leukocytes, possibly facilitating early, pre-symptomatic cancer detection [56,57,58,59]. Tumor-derived cytokines and growth factors (e.g., IL-6, GM-CSF, G-CSF, IL-1β, and VEGF) skew hematopoiesis toward stress myelopoiesis and expansion of immature neutrophil and monocyte populations that acquire myeloid-derived suppressor cell (MDSC) phenotypes. The MDSCs, together with exhausted T cells, blunt T-cell effector function and contribute to systemic T-cell dysfunction (Figure 2) [60,61,62,63].

In parallel, tumor-derived TGF-β, IL-10 and CCL22 recruit and expand regulatory T cells (Tregs), further attenuating Th1/Th17-type responses [64,65,66]. Peripheral blood transcriptomic studies in patients with advanced adenomas or early CRC demonstrate stage-specific systemic immune remodeling, with inflammatory and innate immune networks already altered before clinically overt cancer [57,58]. The Th17/IL-17 axis is crucial for maintaining intestinal epithelial barrier integrity and coordinating neutrophil recruitment to control gut bacteria, so its disruption weakens mucosal containment of opportunistic intestinal bacteria and supports the view of cancer-associated immune dysregulation as a systemic, rather than purely local, immune disorder [67,68]. Tumor-derived cytokines and growth factors (VEGF, G-CSF) further impair conventional type 1 dendritic cell (cDC1) development and function resulting in an immune landscape that is inflamed yet functionally paralyzed [69,70].

These systemic signatures can precede clinical cancer diagnosis by months to years. Circulating hematopoietic stem and progenitor cell compartments show consistent myeloid-biased expansion across several solid cancers [71,72]. Immunotranscriptomic assays can already detect systemic innate immune activation in patients with advanced colorectal adenomas, with emerging T-cell–suppressive signatures that become more pronounced as lesions progress to overt CRC [57,73].

For infectious-disease biology, the consequence is an IL-6/STAT3-inflamed, Th17/IL-17-deficient, DC-poor, and MDSC-expanded immune state, which is permissive for mucosal barrier failure and suboptimal neutrophil antimicrobial function. Minor translocation events from the gut or biliary tree can present as bacteremia, PLA biliary sepsis, or difficult-to-clear pneumonia, illustrating how early-stage cancer functions as a systemic immune disorder that specifically disrupts antibacterial defense.

### 4.2. The TME as a Microbial Incubator

The physical and metabolic state of solid tumors creates permissive microbial niches. Solid tumors are hypoxic and acidic, with abnormal vasculature and disrupted epithelial and endothelial barriers that impair immune cell access and clearance of microbes. These changes alter epithelial antimicrobial programs and tight junction integrity, so barrier function and mucosal containment deteriorate [74,75,76]. Mechanical obstruction of the colon lumen, bile duct, or bronchus produces stasis, bacterial overgrowth, and ascending or post-obstructive infection [77,78]. Intratumoral microbiome studies show that bacteria are not just surface contaminants; they localize intracellularly within tumor and immune cells and form tumor-type–specific communities [79,80]. The same hypoxic, leaky, barrier-disrupted niches that permit this colonization likely also lower the threshold for progression from subclinical colonization to overt infection.

Large registry studies show that bacteremias caused by specific gut taxa are strongly associated with a concurrent or subsequent diagnosis of CRC [9,17,46]. In pancreaticobiliary cancer, acute or recurrent cholangitis frequently reflects malignant biliary obstruction with bile stasis from tumor-induced strictures and may be the first clinical manifestation of an otherwise occult cancer. In lung cancer, ‘non-resolving’ or post-obstructive pneumonia that fails to clear despite appropriate antibiotics is a classic presentation of endobronchial tumor, where airway obstruction and mucus plugging create hypoxic niches that favor local overgrowth and secondary infection [78]. Together, these represent canonical infection phenotypes of cancer-driven ecological failure: colorectal → anaerobe-linked bacteremia; pancreaticobiliary → ascending cholangitis from malignant obstruction; lung → post-obstructive, non-resolving pneumonia.

### 4.3. Dysbiosis, Microbial Virulence Programs, and Translocation Risk

Dysbiosis in colorectal neoplasia is reproducible across cohorts and geography, and the organisms enriched in this state may not be entirely passive. Pks^+^ *E. coli* produces colibactin, generating DNA double-strand breaks; its specific mutational signature has now been detected directly in human CRC genomes and is enriched in early-onset CRC [81,82]. Enterotoxigenic *Bacteroides fragilis* secretes the metalloproteases, cleaves E-cadherin, activates β-catenin and NF-κB signaling, drives epithelial IL-6/STAT3 activation and Th17/IL-17-dependent inflammation and can promote tumorigenesis in experimental models [83,84]. *Fusobacterium nucleatum* binds tumor glycans, inhibits NK and T cell cytotoxicity, activates a TLR4–MYD88–microRNA–autophagy axis [85,86]. In other words, these bacteria do not merely exploit established tumors; they might actively help shape tumor evolution and the immune microenvironment. Importantly, dysbiosis is not unique to CRN: distinct, cancer-type–specific microbial communities have been described in pancreatic ductal adenocarcinoma, cholangiocarcinoma, and lung cancer, including intracellular tumor- and immune-cell–associated bacteria that differ by tumor type and correlate with immune programs, prognosis, and therapy response [80].

Dysbiosis also increases the risk of bacterial translocation. Dysbiotic mucosa shows reduced antimicrobial peptide expression, increased permeability and thinning of the mucus layer [87]. Consequently, the gut-derived organisms that are normally contained (anaerobes, *Enterobacterales*, even some streptococcal groups) are more likely to cross the epithelial barrier. Multiple studies have shown that patients with adenomas and early CRC have a higher prevalence of potentially pathogenic enteric strains, including pathotype-carrying *E. coli* and virulence-encoded *Bacteroides* [88]. This helps explain why a host with occult CRC can develop “spontaneous” anaerobic bacteremia or cryptogenic liver abscess—the mucosal barrier is not intact, and the microbial ecology is shifted toward invasive phenotypes.

### 4.4. Chronic Inflammation as Integrator and Amplifier

Chronic inflammation is the central mechanism that links these processes. NF-κB and IL-6/STAT3 form an inflammatory positive-feedback loop that promotes tumor survival and proliferation, while suppressing antitumor immunity and compromising microbicidal defenses at mucosal barriers [89,90]. Dysbiosis-associated ligands (e.g., LPS, flagellin) activate TLRs to drive NF-κB and IL-6/STAT3 signaling, thereby amplifying tumor fitness, blunting antitumor immunity, altering antimicrobial-peptide programs, and increasing epithelial permeability [91,92]. The net effect suggests that infection can be viewed as an expected clinical phenotype of the developing tumor’s inflammatory architecture.

Taken together, these domains act in series and in parallel; occult tumor induces systemic immune dysfunction; dysbiosis and barrier failure emerge downstream; the TME becomes a microbial incubator; chronic inflammation maintains the malignant state. The clinical infection is simply the visible tip of a multi-year remodeling process driven by tumor evolution. Infection is therefore not always separate from cancer—it can represent one component of the malignant phenotype and, in some patients, the first clinically visible sign of malignancy (Figure 3).

## 5. Clinical Red Flags: Infection Signatures Suggesting Malignancy

For physicians, the key question is not only “what is the source?” but also “why did this infection occur in this host, in this way, at this time?” Large nationwide cohort studies of pneumonia and bloodstream infection show that when an underlying cancer is uncovered after such “sentinel infections”, the excess cancer diagnoses are front-loaded, with cancer incidence peaking in the first 3–6 months after the index episode and then declining thereafter [12,13,15].

Gram-negative bacteremia without an obvious anatomic source is one of the most reproducible infectious “red flags” for occult cancer. In a Danish nationwide cohort of 11,753 adults with Gram-negative bacteremia, the absolute risk of being diagnosed with cancer within 6 months was about 3%, and cancer incidence in that 6-month window was more than threefold higher than expected; over longer follow-up, the overall cancer risk remained about 40% higher than in the general population [13]. Recurrent Gram-negative bacteremia—especially with shifting gut organisms (for example *E. coli* followed by *Klebsiella* and later an anaerobe) in a patient without clear immunodeficiency or a fixed focus—can be interpreted as a pattern suggestive of tumor-driven ecological failure, although this specific recurrence pattern has not yet been systematically quantified in large cohorts.

In practice, infection phenotypes tend to track with the anatomic location of the underlying malignancy. Anaerobic bacteremia, pks^+^ *E. coli*, cryptogenic PLA and even *Fusobacterium* brain abscess cluster with colorectal neoplasia. Recurrent cholangitis without stones or strictures often precedes pancreaticobiliary malignancy, and non-resolving pneumonia is a recognized presenting mode of lung cancer in bronchoscopy series. Recurrent culture-proven UTI in older adults without stones or instrumentation enriches for urothelial carcinoma, and *Clostridium septicum* bacteremia or gas gangrene is a classical signal for occult CRC or hematologic malignancy. *C. difficile* infection (CDI) has also been linked to subsequent CRC diagnosis, and in older adults with recurrent/non-resolving CDI—particularly without recent antibiotics—CRC evaluation could be reasonable [93]. The principal infection–malignancy “red flag” patterns are summarized in Table 2 and illustrated as clinical vignettes in Figure 4.

Laboratory and imaging features increase the probability of cancer: unexplained normocytic anemia, persistent low-grade CRP elevation after microbiologic clearance, cholestatic liver enzymes without biliary explanation, or incidental lymphadenopathy/hepatic hypodensities on CT performed for infection. The major diagnostic error is negative reassurance. A colonoscopy 3–5 years earlier does not exclude CRC. A rapid decrease in CRP after initiation of antibiotic treatment does not exclude malignancy (cancer-associated infections are usually microbiologically responsive). A non-diagnostic CT at the index admission does not exclude cancer (sub-radiologic mucosal or ductal infiltration can drive obstruction, stasis, and overgrowth).

In summary, atypical, recurrent, polymicrobial, or anatomically incongruent infections in adults without conventional risk factors should prompt consideration of both infectious and underlying malignant etiologies.

## 6. From Clue to Diagnosis—“Don’t Miss the Window”

Collectively, these epidemiologic and mechanistic data provide actionable guidance for clinical reasoning. The central principle is that “red-flag infections” which are unusual in presentation, anatomically incongruent, or recurrent without a clear explanation should lower the threshold for targeted cancer evaluation in the anatomically relevant organ system. Therefore, the diagnostic aim after a sentinel infection is not long-term risk prediction, but identification of an already existing malignancy.

Diagnostic evaluation should be parallel and phenotype-directed. Specific organisms and syndromes map to specific anatomic cancer domains, and that mapping is sufficiently reproducible for diagnostic actions to be protocolized. Table 2 summarizes a clinically rational, phenotype-driven diagnostic strategy for the most validated sentinel infections. Simplified, colon phenotypes require colonoscopy, pancreaticobiliary phenotypes require MRCP or pancreas-protocol MRI, lung phenotypes require bronchoscopy—and in high-risk cases, a single cross-sectional CT chest–abdomen–pelvis or FDG-PET/CT should be performed early after the index infection [94,95].

Crucially, the clinical efficiency of this strategy is not theoretical. When recalculated and expressed as estimated number-needed-to-test (NNT, Table 3) to detect one occult malignancy, the magnitudes are comparable to, or higher than, the thresholds used in accepted screening programs. In unselected adults with Gram-negative bacteremia, about 3% are diagnosed with a new cancer within 6 months (NNT ≈ 33) [13]. In bacteremia caused by colorectal-associated anaerobes and in cryptogenic PLA, colorectal neoplasia is found in roughly 3–8% of patients overall (NNT ≈ 12–30) in population-based cohorts and meta-analyses, while East Asian colonoscopy series report colonic neoplasia (cancer or advanced adenoma) in about 15–25% of cryptogenic abscess cases (NNT ≈ 4–7) [9,11,46,96,97]. In *Streptococcus gallolyticus* bacteremia, colorectal neoplasia is detected at colonoscopy in roughly 40–70% of patients (NNT ≈ 2–3), making colon evaluation essentially mandatory [7,98,99]. In non-resolving pneumonia, bronchogenic carcinoma accounts for up to 30% of etiologies overall (NNT ≈ 3–4) [100]. These magnitudes are not negligible; they are similar to accepted screening triggers and importantly, they are obtained without any new screening tool just simply by better interpretation of infections we already see.

A core principle is to balance the benefit of early cancer detection against the risks/costs of over-testing. Most infections do not mean cancer—so the question is not “should we test everyone?” but “who is at sufficiently elevated post-infection risk to justify targeted evaluation?”. Several datasets help narrow this—the excess cancer risk begins to decline after 6 months [12,13]. Therefore, if an initial evaluation after a sentinel infection is negative and the patient remains clinically well for ~12 months, further extensive testing is generally not warranted. Conversely, unusually severe or anatomically unlike infections in younger adults (where background cancer risk is lower) generate a substantially higher SIRs than the same infections in multi-morbid elderly patients, because the event is more biologically aberrant. Clinicians should therefore individualize the threshold for initiating a malignancy evaluation based on age, baseline risk, infection severity, and feasibility of screening tests. These relationships are illustrated through representative clinical vignettes in Figure 4, which highlight when specific infectious presentations warrant targeted cancer evaluation.

Importantly, acting within this window may improve outcomes. Several series report that cancers detected following sentinel infection are more frequently operable. For example, even though a high short-term risk of CRC has been demonstrated in anaerobic bacteremia and detailed comparative data on stage distribution versus usual symptomatic presentation are still limited, CRCs identified after evaluation of cryptogenic PLA are typically nonmetastatic, supporting opportunistic colon investigation in this setting [11,46,97]. In other words, the biology that makes infection an early phenotypic manifestation of cancer also creates a clinical opportunity—if we act on the signal, we may shift diagnosis toward a stage range where cure and long-term control are still possible.

## 7. Future Directions—From Sentinel Infections to Early Cancer Detection

Despite robust associations between selected bacterial infections and subsequent cancer detection, several methodological and interpretive limitations must be recognized. Most studies are observational, often retrospective, and vulnerable to confounding (e.g., smoking and alcohol use as shared drivers of both infection and cancer), detection bias (more imaging after severe infection), and selection bias (hospital-based cohorts enriched for high-risk phenotypes). In particular, patients hospitalized with severe infection undergo more intensive investigations (e.g., imaging, endoscopy), so greater diagnostic intensity likely contributes to the short-term “spikes” in cancer diagnoses after infection. The specificity of individual pathogens is also not absolute—although anaerobic bacteremia strongly points to CRC, similar organisms can also be isolated in advanced diverticular disease or postoperative states. In addition, geography matters; Northern European registries quantify generalizable baseline risks (e.g., ~3% six-month cancer detection after Gram-negative sepsis) [13,15], whereas several East Asian cohorts report larger effect sizes for specific infection–cancer phenotypes. For instance, a Taiwanese population-based study found that prior UTI was associated with a 4–5-fold increased risk of subsequent urinary tract cancers [32]. Some classic infection–cancer links, such as chronic *Salmonella Typhi* carriage and gallbladder carcinoma, are concentrated in typhoid-endemic regions, with meta-analyses and regional studies from South Asia and China showing markedly elevated risks [29,108]. Thus, sentinel infection phenotypes should not be interpreted as stand-alone diagnostic tools, but as context-dependent triggers to lower the threshold for targeted evaluation.

Prospective clinical studies are crucial to advancing the translational momentum in recognizing bacterial infections as early indicators of underlying malignancies. Current efforts include the establishment of prospective cohorts that embed structured cancer evaluation pathways following sentinel infection events, aiming to rigorously quantify the diagnostic yield and clinical utility of infection-based early cancer detection strategies. In parallel, translational research is increasingly focusing on integrating microbial genomic data, host immune profiling, and circulating tumor DNA analyses to develop multi-modal biomarker signatures that can enhance diagnostic precision and specificity. Emerging diagnostics leveraging artificial intelligence to analyze pathogen characteristics—such as species type, recurrence patterns, and infection site incongruence—within comprehensive cancer risk models show promising potential for personalized early detection frameworks. Notably, microbiome-cancer associations are already being explored in cancers such as pancreatic [109], head, neck, and oral malignancies [110], although larger, more robust studies are needed to strengthen their clinical applicability. These efforts underscore a dynamic and growing field that bridges infection biology with oncology, increasing chances for earlier cancer detection and improved patient outcomes.

## 8. Conclusions

Growing epidemiologic evidence, reinforced by modern insights into tumor immunobiology, barrier integrity, and microbiome disruption, shows that certain severe or anatomically unexpected infections can represent early clinical manifestations of occult malignancy. When recognized as sentinel events, these infections provide a practical opportunity for earlier cancer detection, often at stages when curative treatment remains achievable. Although current data are largely observational, the consistency of these associations and the feasibility of focused, organ-specific evaluation support broader clinical application and prospective validation. Emerging approaches, including microbiome and immune profiling and AI-enabled risk stratification, may further enhance recognition of infection-linked cancer signals and expand timely diagnostic opportunities.

## Figures and Tables

**Figure 1 cancers-17-03958-f001:**
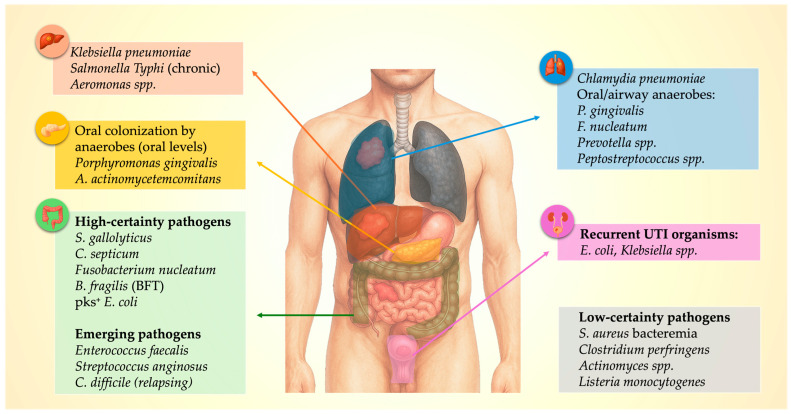
Pathogen–Organ–Cancer Map: Microbial clusters linked to specific sentinel malignancies.

**Figure 2 cancers-17-03958-f002:**
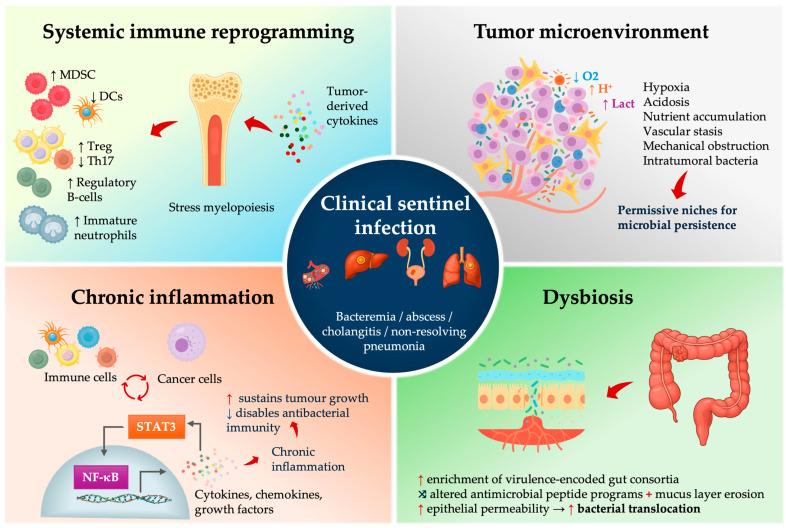
Mechanistic drivers linking occult malignancy to clinical sentinel infection.

**Figure 3 cancers-17-03958-f003:**
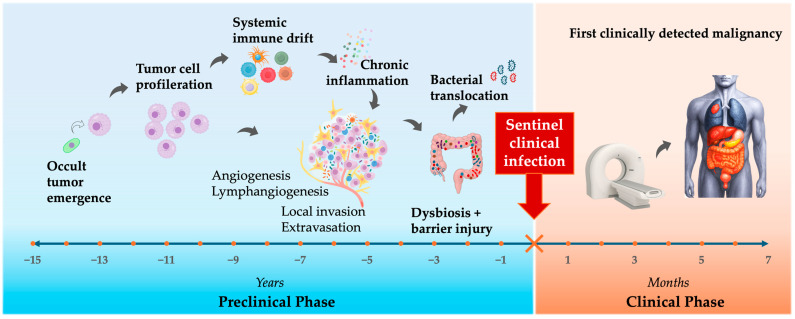
The Sentinel Infection Window: timeline from tumor evolution to infection to cancer detection.

**Figure 4 cancers-17-03958-f004:**
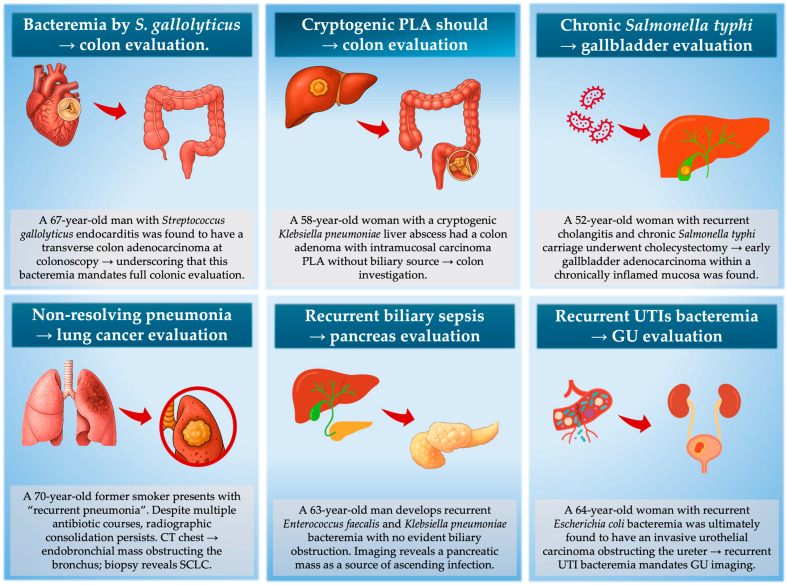
Representative clinical vignettes demonstrating “sentinel infection phenotypes” that should trigger targeted malignancy evaluation. In these scenarios, the infection is not the cause of cancer but the first visible consequence of tumor-driven immune dysfunction, mucosal barrier breakdown, or anatomical obstruction. Acting on these infection signatures enables cancer detection during a narrow diagnostic window.

**Table 2 cancers-17-03958-t002:** Recommended immediate organ-directed evaluation after sentinel infection.

Infection Phenotype	Immediate Targeted Cancer Evaluation
Anaerobic bacteremia (no abscess/perforation)	Colonoscopy (even if 3–5 years prior was normal) ± CT chest–abdomen–pelvis
Cryptogenic pyogenic liver abscess	Colonoscopy + CT chest–abdomen–pelvis ± MRCP if cholestatic enzymes
Recurrent cholangitis without stones/strictures	MRCP or pancreas-protocol MRI (±EUS based on ductal findings)
Non-resolving pneumonia with segmental/lobar collapse	Contrast CT chest (±FDG-PET/CT if available) + diagnostic bronchoscopy
Recurrent bacteremic UTI in older adult (no stones, no instrumentation)	Cystoscopy ± prostate MRI (men) + CT chest–abdomen–pelvis

**Table 3 cancers-17-03958-t003:** Infection phenotypes that should trigger targeted cancer evaluation.

Organ Domain	Sentinel Infection Phenotype	Typical Organisms	Short-Term Cancer Detection Yield */Implied NNT †	Key Sources
Colon	Anaerobic bacteremia (no abscess/perforation)	*Bacteroides*, *Clostridium*, *Fusobacterium*; pks^+^ *E. coli*	~5–10% (≤6–12 mo) → NNT 10–20	[9,46]
	Cryptogenic pyogenic liver abscess	*K. pneumoniae* ± anaerobes	West: ~5–10% → NNT 10–20; East Asia: ~15–30% → NNT 3–7	[10,11,25,101]
	*Fusobacterium nucleatum* brain abscess (no ear–nose–throat or sinus focus of infection)	*F. nucleatum*	case series signal	[102]
	*Clostridium septicum* bacteremia/gas gangrene	*C. septicum*	very high co-occurrence	[46]
Pancreas/biliary	Recurrent cholangitis without stones/strictures	Enterobacterales (esp. *Klebsiella*)	~10% over 5–10 years → NNT ~10	[103]
Liver	Pyogenic liver abscess without risk factors	*Klebsiella*, Enterobacterales	~2% → NNT ~50	[28]
Lung	Non-resolving pneumonia > 4–6 wks despite adequate therapy (lobar collapse/mucus plug)	mixed community-acquired flora; anaerobes in obstruction	~20–30% → NNT 3–5	[104,105]
Urinary	Recurrent bacteremic UTI in older adult (esp. male or post-menopausal; no stones/instrumentation)	*E. coli*, *Klebsiella*	OR ~2–7; not well quantified	[33,35]
Systemic	Cryptogenic Gram-negative bacteremia (overall)	*E. coli*, *Klebsiella*, others	~3% (≤6 mo) → NNT ≈ 33	[13,15]
Fever of unknown origin	≥50 y with persistent inflammatory signature	—	~10–20% → NNT 5–10	[106,107]

* short-term = typically ≤6–12 months after index infection; † NNT = approximate number needed to test to detect 1 cancer (NNT values were calculated as the inverse of absolute cancer detection percentage within the published time window).

## Data Availability

No new data were created or analyzed in this study. Data sharing is not applicable to this article.

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
