# Peer review of "The Infectious Clue: Linking Bacterial Infections to Underlying Malignancies"

_cancers, 2025, doi:10.3390/cancers17243958_

Round 1
Reviewer 1 Report
Comments and Suggestions for Authors
This is overall a good article on a very timely topic. I congratulate the authors for the excellent work! The intersection between ID and cancers is indeed worthy of being explored.
Just some minor comments below that could perhaps help improve clarify for an outside reader:
- Some sections explore the link between infection and risk of cancer development, i.e., chronic S. Typhi carriage and pancreaticobiliary cancers (section 2.2), Chlamydia and lung cancers (section 2.3), whereas other focus on infections as unmasking signs for cancers. For unity, it would be important to follow the same pattern in reporting across all sections, for example start each section with infections increasing the risk of cancer development, and follow with infections unmasking cancers.
- Table 1 is extremely useful and very informative.
- Figure 1 is also very good, but perhaps one aspect could be clarified: The oral anaerobes and pancreatic cancer – it should be clarified on the figure that this refers to oral levels of these pathogens (as described in the text on lines 134-135), and not pancreatic infection with those germs per se.
- Spikes in malignancies following infection should be put into context, as this can also be due to healthcare-seeking behavior and the higher likelihood of getting a diagnostic imaging during hospitalization for infection, than in a community-dwelling person with no hospital contact.
- Figure 3 is also interesting. I would suggest revising the term “clinically-detectable” to “clinically detected”, since perhaps the malignancy could have been detected earlier if diagnostic testing had been performed.
Hope these suggestions are useful and again, congratulations to the authors for an excellent review of this important topic.
Author Response
REVIEWER 1
This is overall a good article on a very timely topic. I congratulate the authors for the excellent work! The intersection between ID and cancers is indeed worthy of being explored. Just some minor comments below that could perhaps help improve clarify for an outside reader:
AUTHORS’ ANSWER: Thank you for the positive feedback and constructive suggestions. All points have been addressed as detailed below, with tracked changes in the revised manuscript.
- Some sections explore the link between infection and risk of cancer development, i.e., chronic S. Typhi carriage and pancreaticobiliary cancers (section 2.2), Chlamydia and lung cancers (section 2.3), whereas other focus on infections as unmasking signs for cancers. For unity, it would be important to follow the same pattern in reporting across all sections, for example start each section with infections increasing the risk of cancer development, and follow with infections unmasking cancers.
AUTHORS’ ANSWER: We thank the reviewer for this helpful comment. We have added a brief clarification at the beginning of Section 2. Sections 2.1–2.5 now follow a consistent structure, with each subsection first describing short-term infections that “unmask” occult cancer and then summarizing long-latency infections that may promote carcinogenesis.
- Table 1 is extremely useful and very informative.
- Figure 1 is also very good, but perhaps one aspect could be clarified: The oral anaerobes and pancreatic cancer – it should be clarified on the figure that this refers to oral levels of these pathogens (as described in the text on lines 134-135), and not pancreatic infection with those germs per se.
AUTHORS’ ANSWER: We thank the reviewer for this helpful comment. In Figure 1, we have clarified that the link between oral anaerobes and pancreatic cancer refers to oral colonization levels rather than pancreatic infection.
- Spikes in malignancies following infection should be put into context, as this can also be due to healthcare-seeking behavior and the higher likelihood of getting a diagnostic imaging during hospitalization for infection, than in a community-dwelling person with no hospital contact.
AUTHORS’ ANSWER: We thank the reviewer for this important comment. We have now more explicitly acknowledged the role of healthcare-seeking behavior and diagnostic intensity in interpreting short-term “spikes” in cancer diagnoses after infection. In Section 2, we added a brief sentence noting that greater healthcare contact and more intensive investigations around severe infection likely contribute to the early excess in cancer diagnoses. In Section 7 (Future Directions), we expanded the discussion of methodological limitations, including detection bias due to increased diagnostic intensity, while emphasizing that this alone is unlikely to fully explain the observed pathogen- and organ-specific patterns.
- Figure 3 is also interesting. I would suggest revising the term “clinically-detectable” to “clinically detected”, since perhaps the malignancy could have been detected earlier if diagnostic testing had been performed.
AUTHORS’ ANSWER: We have updated Figure 3 as suggested, replacing “clinically-detectable” with “clinically detected”.
Reviewer 2 Report
Comments and Suggestions for Authors
Overall, this is a well-written and comprehensive review that addresses an important and clinically relevant topic. The manuscript is thoughtfully structured, the literature coverage is impressive, and the proposed clinical framework is highly valuable. To further strengthen the work, I would suggest considering the following points:
-
Section 4 is too long, and several mechanistic details feel unnecessary for a clinically focused review. A more concise version would improve clarity.
-
The strength of evidence is not always balanced. Well-established links (e.g., S. gallolyticus and colorectal cancer) are presented at a similar level as associations that are only anecdotal. These should be clearly differentiated.
-
Some sentences sound too close to implying causation. For example, the dysbiosis → cancer progression discussion should use more cautious wording to avoid overstating the evidence.
-
Methodological limitations need more emphasis. Confounding, detection bias, and surveillance bias should be highlighted earlier, so readers understand that most data come from retrospective observational studies.
Author Response
REVIEWER 2
Overall, this is a well-written and comprehensive review that addresses an important and clinically relevant topic. The manuscript is thoughtfully structured, the literature coverage is impressive, and the proposed clinical framework is highly valuable. To further strengthen the work, I would suggest considering the following points:
- Section 4 is too long, and several mechanistic details feel unnecessary for a clinically focused review. A more concise version would improve clarity.
AUTHORS’ ANSWER: We thank the reviewer for this comment. We agree that a more concise mechanistic section improves readability for a clinically focused audience. We therefore streamlined Section 4 by condensing several highly detailed pathway descriptions (e.g., specific cytokine and signaling cascades in Sections 4.1, 4.2 and 4.3) and merging overlapping explanations of dysbiosis and chronic inflammation, while preserving the main mechanistic concepts that are directly relevant to the clinical framework.
- The strength of evidence is not always balanced. Well-established links (e.g., gallolyticusand colorectal cancer) are presented at a similar level as associations that are only anecdotal. These should be clearly differentiated.
AUTHORS’ ANSWER: We thank the reviewer for this important comment and agree that the strength of evidence should be more clearly differentiated. We have now explicitly stratified infection–cancer links by evidence level. In the introductory part of Section 2 we added a sentence explaining that we distinguish high-certainty, emerging and low-certainty/anecdotal associations throughout the review. In Section 2.1 we now refer to S. gallolyticus and C. septicum as high-certainty sentinel pathogens for CRC, whereas in Section 2.5 we explicitly label associations involving S. aureus, Clostridium perfringens, Actinomyces spp. and Listeria monocytogenes as low-certainty and largely anecdotal.
- Some sentences sound too close to implying causation. For example, the dysbiosis → cancer progression discussion should use more cautious wording to avoid overstating the evidence.
AUTHORS’ ANSWER: We have carefully re-read the entire manuscript and revised wording where it might unintentionally imply definitive causality, particularly in the dysbiosis → cancer progression section. We now use more cautious language (e.g., “may,” “can,” “in experimental models,” “plausible explanation”) and explicitly emphasize the predominantly observational nature of the evidence.
- Methodological limitations need more emphasis. Confounding, detection bias, and surveillance bias should be highlighted earlier, so readers understand that most data come from retrospective observational studies.
AUTHORS’ ANSWER: We thank the reviewer for this important comment. We have now emphasized the methodological limitations earlier in the manuscript. In Section 2, immediately before the organ-specific subsections, we added a sentence stating that most data derive from retrospective observational studies (registry-based cohorts and case–control designs) and are subject to confounding, detection bias and surveillance bias, with a pointer to the detailed discussion in Section 7.
Reviewer 3 Report
Comments and Suggestions for Authors
The authors present a very well written review on an important topic which does not receive appropriate attention in practice or literature.
The manuscript does not need a lot of improvement. However, the authors should separate more clearly between sentinel infections for existing cancer and infections that cause cancer (e.g. Helicobacter, Salmonella)
Author Response
REVIEWER 3
The authors present a very well written review on an important topic which does not receive appropriate attention in practice or literature.
The manuscript does not need a lot of improvement. However, the authors should separate more clearly between sentinel infections for existing cancer and infections that cause cancer (e.g. Helicobacter, Salmonella)
AUTHORS’ ANSWER: We thank the reviewer for the positive assessment and for this helpful suggestion. We have now more clearly distinguished between infections that act as sentinel events for existing, occult cancer and infections that may contribute to cancer development over time. In Section 2, we explicitly introduce this distinction and structure the organ-specific subsections so that sentinel infections (e.g., bacteremia, pneumonia, cholangitis) are described first, followed by long-latency carcinogenic infections such as Helicobacter pylori and chronic Salmonella Typhi carriage. We also use this terminology consistently throughout the manuscript.